# Mechanism of microtubule stabilization by taccalonolide AJ

Yuxi Wang[1,2], Yamei Yu[1], Guo-Bo Li[1], Shu-Ang Li[1], Chengyong Wu[1], Benoît Gigant[3], Wenming Qin[4], Hao Chen[1], Yangping Wu[1], Qiang Chen[1] & Jinliang Yang[1]

As a major component of the cytoskeleton, microtubules consist of αβ-tubulin heterodimers and have been recognized as attractive targets for cancer chemotherapy. Microtubule-stabilizing agents (MSAs) promote polymerization of tubulin and stabilize the polymer, preventing depolymerization. The molecular mechanisms by which MSAs stabilize microtubules remain elusive. Here we report a 2.05 Å crystal structure of tubulin complexed with taccalonolide AJ, a newly identified taxane-site MSA. Taccalonolide AJ covalently binds to β-tubulin D226. On AJ binding, the M-loop undergoes a conformational shift to facilitate tubulin polymerization. In this tubulin–AJ complex, the E-site of tubulin is occupied by GTP rather than GDP. Biochemical analyses confirm that AJ inhibits the hydrolysis of the E-site GTP. Thus, we propose that the β-tubulin E-site is locked into a GTP-preferred status by AJ binding. Our results provide experimental evidence for the connection between MSA binding and tubulin nucleotide state, and will help design new MSAs to overcome taxane resistance.

[1] State Key Laboratory of Biotherapy and Cancer Center, West China Hospital, Sichuan University, and Collaborative Innovation Center of Biotherapy, Chengdu 610041, China. [2] Department of Respiratory Medicine, West China Hospital of Sichuan University, Chengdu 610041, China. [3] Institute for Integrative Biology of the Cell (I2BC), CEA, CNRS, Univ. Paris-Sud, Université Paris-Saclay, Gif-sur-Yvette 91198, France. [4] National Center for Protein Science Shanghai, Institute of Biochemistry and Cell Biology, Chinese Academy of Sciences, Shanghai 201210, China. Correspondence and requests for materials should be addressed to Q.C. (email: qiang_chen@scu.edu.cn) or to J.Y. (email: jinliangyang@scu.edu.cn).

Microtubules are cytoskeletal filaments that play essential roles in cell trafficking, signalling, migration and division, and have been treated as important targets for cancer drugs[1]. Microtubules are dynamic structures consisting of αβ-tubulin heterodimers that assemble into protofilaments in a head-to-tail fashion, and the straight and parallel protofilaments interact laterally to form the microtubule hollow cylinder. Tubulin heterodimers contain two GTP-binding sites, termed N-site (non-exchangeable) and E-site (exchangeable). The GTP bound to the α-tubulin N-site is stable and plays a structural function. However, the GTP bound to the β-tubulin E-site may be hydrolysed to GDP shortly after assembly. GTP bound at the E-site makes tubulin dimer more prone to polymerization, while tubulin dimer with GDP bound at the E-site tends to depolymerize. This GTP cycle is thus essential for the dynamic instability of the microtubule.

A wide range of agents bind to tubulin and interfere with microtubule function leading to cellular death: some agents inhibit assembly, while others inhibit disassembly. Microtubule-stabilizing agents (MSAs) promote polymerization of tubulin and stabilize the polymer, preventing depolymerization. The microtubule stabilizer paclitaxel (Taxol) and its second-generation analogue docetaxel (Taxotere) are successfully used in the clinic for the treatment of various cancers[2]. The significant anticancer activities of these taxanes make the microtubule stabilizers particularly interesting in cancer therapeutics[2]. However, the limitations of their intrinsic and acquired drug resistance and dose-limiting toxicities prompt the development of new classes of microtubule-stabilizing drugs[3,4].

The taccalonolides, a newly identified class of MSAs isolated from plants of the genus *Tacca*, have a distinct structure and microtubule-stabilizing manner compared to other microtubule stabilizers, and, most importantly, have shown the ability to circumvent taxane resistance[5]. The taccalonolides overcome drug resistance mediated by mutations in the taxane-binding site, the overexpression of multidrug resistance protein 7, P-glycoprotein and the βIII tubulin isotype *in vitro*, and show excellent *in vivo* antitumour activity in a paclitaxel- and doxorubicin-resistant murine tumour model[6,7]. These properties make the taccalonolides attractive next-generation MSAs for chemotherapy.

Taccalonolides A and E were newly isolated and identified as microtubule stabilizers[8]. However, neither direct interaction with microtubules nor enhancement of the polymerization of purified tubulin was detected for taccalonolides A and E[9]. Identified later, the significantly more potent taccalonolides AF and AJ demonstrated for the first time a direct interaction of the taccalonolides with purified tubulin[10]. Taccalonolide AJ exhibited a high potency with a half-maximal inhibitory concentration value of 4.2 nM, comparable to paclitaxel[10]. It has been demonstrated that the taccalonolides AF and AJ covalently bind to microtubules[11]. However, the exact residue involved in AF or AJ covalent binding has not been determined.

Here we determine the 2.05 Å crystal structure of the AJ–tubulin complex. The structure reveals that AJ C22–C23 epoxide group covalently binds to β-tubulin D226. Based on the crystal structure, nuclear magnetic resonance (NMR) information and computational analysis, we propose a reaction mechanism for the covalent bond formation between AJ and β-tubulin D226. On AJ binding, the M-loop undergoes a closed-to-open conformational shift and forms a short helix. Surprisingly, we have found that the E-site in β-tubulin is occupied by GTP instead of GDP, suggesting that, on AJ binding, β-tubulin adopts a conformation prone to retain GTP. Biochemical assays confirm that AJ is able to inhibit the hydrolysis of the E-site GTP. Thus, we propose that the β-tubulin E-site is locked into a GTP-preferred status by AJ binding. Our studies provide

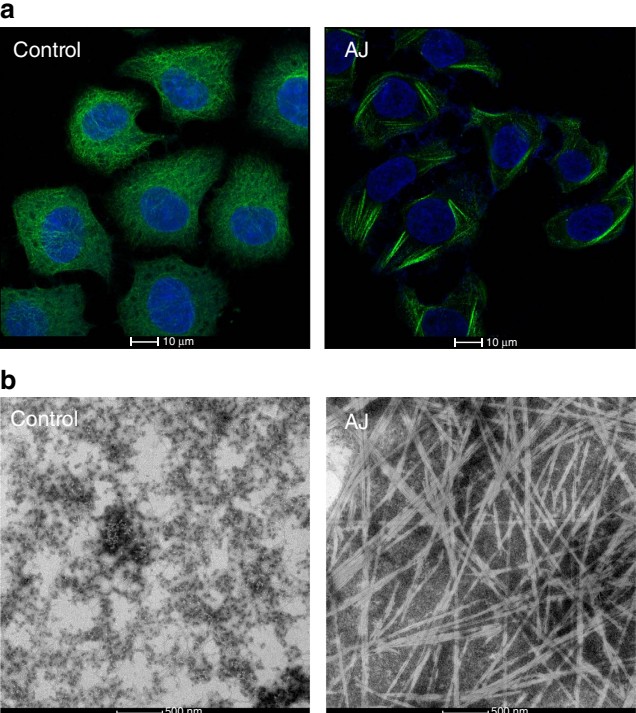

**Figure 1 | Effect of AJ on tubulin oligomerization.** (**a**) Immunofluorescence confocal microscopy images of HepG2 cells treated with DMSO (control) or AJ. The nuclei and microtubules have been labelled with DAPI (blue) and α-tubulin antibody (green), respectively. (**b**) Transmission electron micrographs of negatively stained tubulin specimens obtained in the absence (control) and presence of AJ.

insights into the microtubule-stabilizing mechanism of the taccalonolides AJ as compared to other MSAs, and will be useful for the design of new MSAs to overcome taxane resistance.

## Results

**Effect of AJ on *ex vivo* and *in vitro* microtubule assembly.** We assessed the activity of taccalonolide AJ on microtubule assembly using immunofluorescence and electron microscopy to monitor tubulin oligomer formation. AJ induced the formation of bundle-like tubulin oligomers in HepG2 cells (Fig. 1a), and stimulated the polymerization of purified tubulin (Fig. 1b).

**AJ binds covalently to β-tubulin D226.** Several compounds have been shown to bind covalently to tubulin, such as zampanolide (Zampa) to βH229 (ref. 12) and pironetin to αC316 (refs 13,14). AJ has also been suggested to react covalently with tubulin and the binding region has been narrowed down by mass spectrometry to the β212–230 peptide[11]. To determine the exact residue on β-tubulin to which AJ binds covalently, we used crystals of a protein complex composed of two αβ-tubulin heterodimers, the stathmin-like protein RB3 and tubulin tyrosine ligase (T2R–TTL)[12,15], and solved the tubulin-bound AJ structure at 2.05 Å resolution (Fig. 2). The electron density of AJ in this crystal structure is well defined and allows us to determine the orientation and conformation of the ligand unambiguously (Fig. 2c).

The taccalonolide AJ bound within the taxane site, which is located at the microtubule lumen. AJ formed hydrogen bonds with β-tubulin D226, H229, T276 and R278, and a weak hydrogen bond with K19 (3.7 Å) (Fig. 2a). The tubulin residue

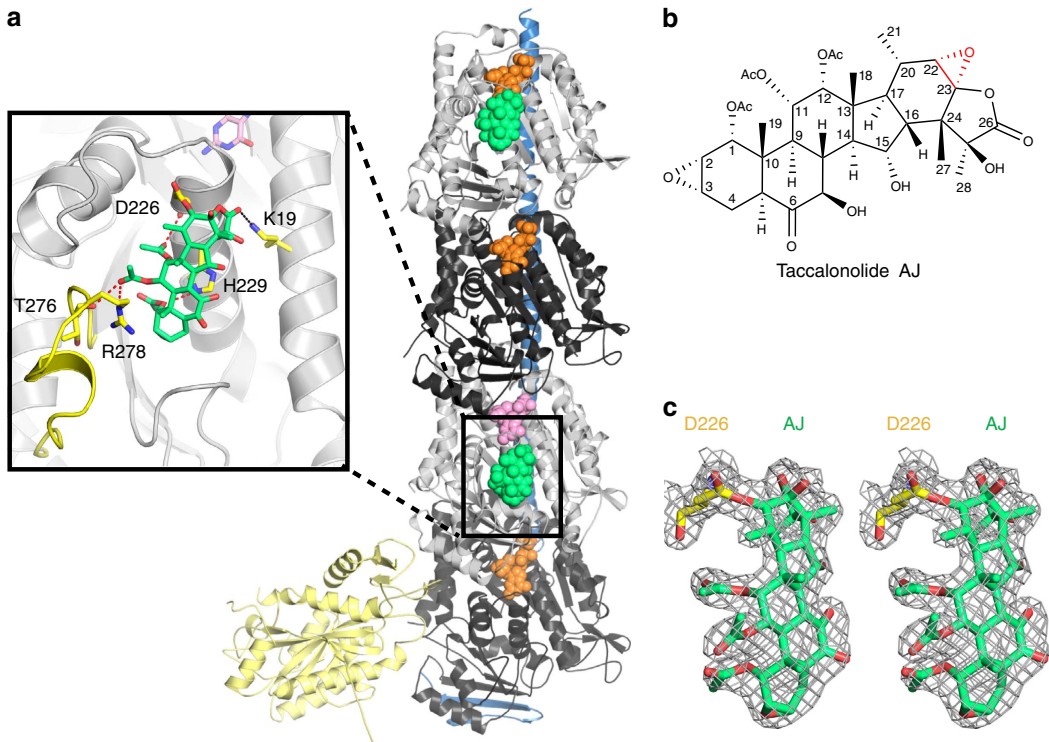

**Figure 2 | AJ covalent binding to β-tubulin.** (**a**) Overall structure of the complex crystallized. The RB3-SLD is coloured blue; TTL, light yellow; α-tubulin, black; β-tubulin, grey; GTP, orange; GDP, pink and AJ, green. The details of AJ binding are enlarged and the β M-loop is highlighted in yellow. (**b**) Chemical structure of taccalonolide AJ. The epoxide group between C22 and C23 is highlighted in red. (**c**) Stereo view of the Fo–Fc omit map (contoured at 3σ) of AJ and the side chain of βD226, clearly showing the covalent bond between them.

numbering is according to Löwe et al.[16]. Most importantly, using its C22–C23 epoxide group, AJ covalently bound to the carboxyl group of βD226 side chain (Fig. 2a), consistent with the reported mass spectrometry data[11]. Taccalonolides AF and AJ are generated from their parent taccalonolides A and B, respectively, by epoxidizing the C22–C23 double bond to an epoxide group. This simple epoxidation dramatically increased the potency over 200- and 700-fold, respectively[10]. The finding that AJ covalently binds to βD226 readily explained why AF and AJ, instead of other taccalonolides that do not contain the C22–C23 epoxide group, could bind tubulin directly and showed low nanomolar potency against cancer cell lines[10].

Another taccalonolide analogue, AI epoxide, reported recently by Peng et al.[17], shows an even higher potency than AJ[17]. We generated a docking model for β-tubulin complexed with AI epoxide (Supplementary Fig. 1), which showed that AI epoxide might have a similar binding mode to that of AJ. The bulky isovaleryloxy group at C-1 of AI epoxide is well positioned into a hydrophobic pocket surrounded by L217, L230, L275 and F272, which may be the reason for its higher potency than AJ.

**Reaction mechanism of AJ covalent binding to β-tubulin.** For the covalent linkage between AJ C22–C23 epoxide group and the carboxyl group of βD226, SN2 reaction is the most common type. However, the chirality of this epoxide group we observed in the crystal structure is not consistent with the SN2 reaction mechanism (Supplementary Fig. 2).

To figure out the chirality contradiction, we first inspected the proton NMR spectrum of AJ. The chemical shift of H22 at 3.253 p.p.m., which is at 5.00 p.p.m. in taccalonolide B[10], confirmed an epoxy group at C22–C23 (Supplementary Fig. 3).

The NOESY (Nuclear Overhauser Effect SpectroscopY) results showed a strong NOE correlation between H21 and H22 (Supplementary Fig. 4), indicating a close proximity between these two H atoms. We have attempted to obtain the crystal structure of taccalonolide AJ but failed. We then carried out Gaussian calculations to compare each bond angle size and distance between the protons for taccalonolide AJ (α) and AJ (β) (Supplementary Fig. 5a). The dihedral angles between H21 and H22 for α or β configuration are 71° and 111°, respectively, both have a small coupling constant (0–4 Hz) and thus could not be used to determine the epoxide moiety configuration. The distances between H21 and H22 for α or β configuration are 2.46 and 2.50 Å, respectively, indicating that the NOE correlation signal between H21 and H22 is indistinguishable for α and β configurations. Finally, we found that the distances between H22 and H27 or H28 show a large difference (for the α configuration, the distance between H22 and H27 is smaller than that between H22 and H28; while for the β configuration, the distance between H22 and H27 is larger than that between H22 and H28) (Supplementary Fig. 5a). NOE data show that the signal strength between H22 and H27 is eight times higher than that between H22 and H28 (Supplementary Fig. 5b), which supports that the C22–C23 epoxide of AJ is in α configuration. Owing to the α configuration of AJ, it is unlikely for the tubulin–AJ ester linkage to form by an SN2 reaction. We then propose another reaction mechanism for it, in which the activation of AJ C22–C23 epoxide by protonation is followed by two interesterification processes, eventually leading to the relative stable tubulin–AJ ester bond (Fig. 3). This covalent reaction mechanism, which is more complicated than SN2 reaction, may provide an explanation for the slow rate of tubulin polymerization initiated by AJ[11].

**Figure 3 | Possible reaction mechanisms of the ester bond between taccalonolide AJ and β-tubulin D226.** When AJ binds to the taxane site, the epoxy ring between C22 and C23 approximates the carboxyl group of βD226 and is activated by protonation; then, C23 is attacked by the carboxylate of D226 resulting in the formation of the oxirane ring-opening hydroxyl intermediate **3**, followed by an interesterification to form intermediate **4**. The five-membered ring of intermediate **4** has two adjacent carbons on the six-membered ring in *trans* position, in which the ring strain is relatively large, thus driving to proceed another interesterification to form the complex **5**, as observed in the crystal structure.

**AJ binding induces conformational changes of the β M-loop.** The M-loop is a central element for lateral tubulin contacts between protofilaments in microtubules[18,19]. In α-tubulin, the position corresponding to the taxane site on β-tubulin is occupied by the extra eight residues inserted into the S9–S10 loop. The taxane-binding site MSAs may mimic these extra residues to stabilize the conformation of the M-loop. The M-loop of β-tubulin is disordered in some tubulin crystal structures (PDB IDs 3HKB, 3HKC, 4O2A and 4O2B), representing its flexibility; while in the recently solved tubulin–destabilizer complex structures (PDB IDs 5CA0, 5CA1, 5C8Y and 5CB4), the M-loop of β-tubulin is ordered and adopts a closed conformation (Fig. 4a). In the tubulin–AJ structure reported here, the covalently bound AJ pushes the M-loop outward to form an open conformation (Fig. 4a). This closed-to-open flipping positions the M-loop in a conformation prone to form lateral tubulin interactions and thus to promote microtubule assembly.

Another conformational change of the β-tubulin M-loop is that it forms a short helix, similar to that in the tubulin–Zampa and tubulin–EpoA complex structures[12] (Fig. 4b). Such conformation change of the M-loop has been proposed to facilitate the lateral interactions in microtubules[12].

These observations in our structure support that AJ induces the closed-to-open and loop-to-helix conformational changes of the β-tubulin M-loop to facilitate establishing lateral tubulin contacts in microtubules.

**AJ regulates the tubulin nucleotide state.** Compared to the two tubulin–MSA complex structures reported previously[12], the tubulin–AJ complex presented two differences. First, AJ was bound to both β-tubulin subunits in the T2R–TTL complex, while Zampa or EpoA only occupied one binding site. Second, a GTP molecule instead of GDP was found in the E-site of one β-subunit (Fig. 4c,d). This E-site conformation in the AJ–tubulin structure is similar to that of GTP–tubulin (PDB ID 3RYF), and differs from Zampa– or EpoA–tubulin structures mainly in D179 and H2 (Fig. 4c). This striking finding suggests that AJ binding may lock the β-tubulin E-site into a conformation prone to bind GTP and inhibit GTP hydrolysis. However, GTP-bound E-site has also been found in a crystal obtained by

seeding[20], where the crystal was collected 8 h after the crystallization set-up. Then, we used reverse-phase HPLC to monitor the hydrolysis of E-site GTP on unassembled tubulins with or without the presence of AJ, and confirmed that AJ indeed inhibits GTP hydrolysis of tubulin (Fig. 4e,f).

Our structural and functional data suggest that AJ binding locks the interface of β-tubulin into a GTP-binding-preferred conformation and inhibits the hydrolysis of the E-site GTP. This may result in more stable soluble GTP–tubulin molecules, which promote the assembly of microtubules. In addition, it might delay GTP hydrolysis after assembly, resulting in a more stable GTP-cap at microtubule ends, therefore reinforcing microtubule stability.

**The microtubule-stabilizing mechanism of AJ.** Thus, the binding of AJ may stabilize microtubule in three ways: (1) AJ binding induces a closed-to-open and loop-to-helix conformational change of β M-loop, which may facilitate establishing lateral tubulin contacts in microtubules; (2) AJ may promote tubulin polymerization by binding to soluble tubulin and stabilizing it into the assembly-competent GTP state; (3) AJ inhibits microtubule depolymerization by stabilizing the GTP-cap at the (+) tip of a microtubule. The microtubule-stabilizing mechanism of AJ has been illustrated in Fig. 4g. Our model supports that AJ affects both the lateral and longitudinal contacts of microtubule. This mechanism is in line with the recent reports that MSAs induce structuring of the M-loop into a short helix to promote microtubule assembly and stabilization[12], and MSAs allosterically affect remodelling of the longitudinal interdimer interface[19].

**Discussion**

The taccalonolides are new taxane-site microtubule stabilizers, showing the ability to circumvent taxane resistance[5]. We present a 2.05 Å crystal structure of tubulin complexed with taccalonolide AJ, which covalently binds to β-tubulin D226 via its C22–C23 epoxide group. Using various approaches, we exclude the possibility of SN2 mechanism for the tubulin–AJ ester linkage, and propose another reaction mechanism for it.

On AJ binding, the β-tubulin M-loop undergoes a closed-to-open shift and a loop-to-helix conformational change. A similar

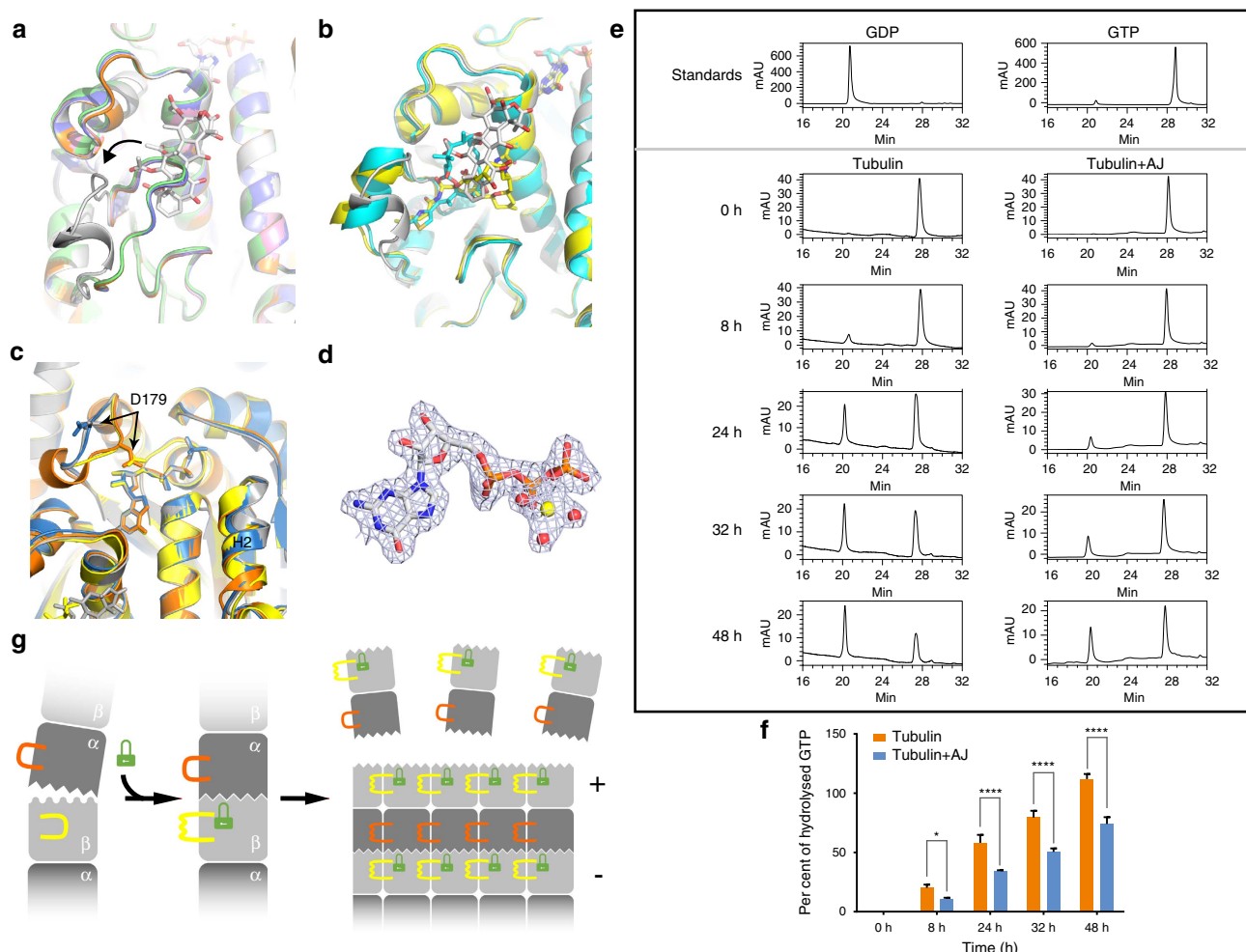

**Figure 4 | The microtubule-stabilizing mechanism of AJ.** (**a**) Superposition of tubulin–AJ (grey) and taxane-site-free tubulin structures (5CA0: pink; 5CA1: orange; 5C8Y: blue; 5CB4: green) shows the closed-to-open conformational change of the β M-loop. (**b**) Superposition of tubulin–MSA structures (AJ: grey; Zampa: yellow; EpoA: cyan) shows the helix conformation of the β M-loop. (**c**) Comparison of E-site conformations. The β-subunits of AJ-bound tubulin (grey), GTP-preferred status tubulin (3RYF: blue), Zampa-bound tubulin (orange) and EpoA-bound tubulin (yellow). (**d**) Electron density of GTP at the E-site in tubulin–AJ complex, with a magnesium ion (yellow) and three water molecules (red). (**e**) Reverse-phase HPLC (RP-HPLC) assay shows that AJ binding significantly delayed E-site GTP hydrolysis. (**f**) Quantification of the hydrolysis of the E-site GTP. The area of GDP and GTP peaks in **e** are calculated. Values are means ± s.d. ($n = 3$). *$P < 0.05$; ****$P < 0.0001$. (**g**) Proposed microtubule-stabilizing mechanism of AJ. Binding of AJ (green lock) to the taxane-site induces the β M-loop (yellow) to undergo a closed-to-open and loop-to-helix conformational change, and locks the E-site into a GTP-preferred conformation (sawtooth). The GDP-status conformation is shown as wave.

conformational transition has also been found in HGPRT, a key enzyme in the salvage pathways for purine nucleotide synthesis[21]. The flexible loop (loop II) of HGPRT, playing a critical role in catalysis, is usually disordered in crystal structures without the presence of substrate. However, in a thermophilic HGPRT, this flexible loop adopts a half-closed and helical conformation that has been proposed to facilitate the catalytic reaction[21].

However, a recent report shows that no significant differences in the lateral interfaces have been observed between the GMPCPP, GDP and GDP-Taxol electron microscopy structures, arguing against that the M-loop conformation plays an essential role in taxane site stabilization mechanism[19]. We propose that this discrepancy is due to the fact that the specimens in the latter study are all in polymerized status, and thus the M-loop conformation is supposed to be constrained by the microtubule lattice.

Compared to AJ, the previously reported MSAs Zampa and EpoA occupy a position closer to the M-loop and induce it to form a longer helix (Fig. 4b). This suggests that AJ has a lesser effect on the M-loop than Zampa or EpoA. Consistently, previous results of hydrogen/deuterium exchange mass spectrometry show that docetaxel stabilizes the M-loop of β-tubulin, but AJ does not[11]. The stabilizing effect of AJ may be beyond the detection limit of the hydrogen/deuterium exchange mass spectrometry method.

In the tubulin–AJ complex structure, a GTP instead of GDP molecule occupies the E-site of one β-subunit (Fig. 4c,d). We then use biochemical assays to confirm that AJ indeed inhibits the hydrolysis of tubulin-bound GTP. AJ can enhance the rate and extent of tubulin polymerization, and the resulting microtubules are profoundly cold stable[11]. These effects resemble those of the non-hydrolysable GTP analogues GMPPNP and GMPCPP[22,23], which is consistent with the inhibiting effect of AJ on E-site GTP hydrolysis.

The stochastic switch between growing and shrinking of the microtubule, referred to as dynamic instability, is driven by GTP hydrolysis[24]. Tubulin adds onto the end of the microtubule in the GTP-bound state, while GDP–tubulin is more prone to

depolymerization[25]. A cap of GTP-bound tubulin is proposed to exist at the tip of the microtubule, protecting it from disassembly, while a GDP-bound tubulin subunit at the (+) tip of a microtubule will tend to fall off[26]. The inhibition of the tubulin-bound GTP hydrolysis by AJ may have two consequences: (1) an increase of the GTP–tubulin concentration, and (2) the stabilization of the first layer of GTP-cap, which mimics the soluble tubulin. These endow AJ with the ability to promote the assembly of microtubule and stabilize microtubule from disassembly.

There are two β-subunits in the T2R–TTL complex. In the structure we determined, only one E-site is occupied by GTP (Fig. 2a). Tubulin itself serves as GAP (GTPase-activating protein) for β-tubulin of the adjacent heterodimer in a protofilament. Subunit addition at the plus end significantly promotes hydrolysis of GTP bound to the now interior β-tubulin. The β-tubulin within a microtubule cannot exchange its bound GDP for GTP. In the T2R-TTL complex, the middle β-subunit (chain B) is in a situation mimicking the β-tubulin within a curved protofilament (Fig. 2a), thus hydrolysis of GTP at this site is significantly accelerated and likely has happened before we add AJ into the crystals. The resulting GDP is not able to be released, therefore we observe a GDP in the middle β-subunit (chain B). In contrast, the hydrolysis of the E-site GTP in the top β-subunit (chain D) is much slower, and could be exchanged to a new GTP after hydrolysis. In any case, on AJ binding, the hydrolysis of GTP at this site will be inhibited.

To evaluate the clinical usefulness of the taccalonolides, supply and formulation issues must be considered. Owing to the structural complexity of these compounds, the complete chemical synthesis for large-scale supply seemed cost prohibitive even if it were achieved. Because the taccalonolides are less water soluble than paclitaxel[27], the development of a non-toxic, readily bioavailable formulation is another challenge. Our crystal structure revealed that the taccalonolide AJ covalently bound to D226 of β-tubulin using its C22–C23 epoxide group. It is possible to design new compounds with an epoxide group or add an epoxide moiety to MSAs that bind within the taxane site to develop better anticancer drugs.

## Methods

**Immunofluorescence.** HepG2 cells were obtained from the Cell Bank of the Chinese Academy of Sciences (Shanghai, China). Mycoplasma contamination of cell line were tested via Hoechst33258 staining, and no contamination was observed. HepG2 cells were grown on glass coverslips in 6-well plates (at a density of $1 \times 10^6$ per well) overnight in a 5% $CO_2$ incubator at 37 °C and then treated with 30 nM taccalonolide AJ (solubilized in dimethylsulfoxide (DMSO)) or same amount of DMSO as control for 16 h. After treatment, cells were fixed with 4% paraformaldehyde and then washed two times with PBS, and then treated with PBS containing 0.5% Triton X-100. After treatment with blocking buffer (PBST containing 5% BSA) for 1 h at room temperature, microtubules were marked using anti-α-tubulin (Santa Cruz; sc-5286) antibody (1:100 dilution in blocking buffer). Then, cells were incubated overnight at 4 °C. After rinsing three times with PBST, cells were stained with FITC-conjugated fluorescence second antibody (Proteintech; SA00003-1) and labelled with 0.1 g ml[−1] 4,6-diamidino-2-phenylindole (Beyotime; C1002) and then washed five times with PBST. Samples were visualized and photographed using a Zeiss LSM 880 with Airyscan laser confocal microscope (laser 405 nm, 488 nm).

**Transmission electron microscopy.** Taccalonolide AJ (10 mM in DMSO) was added to tubulin (50 μl, 1 mg ml[−1]) to a final concentration of 10 μM. Solution was incubated at room temperature for 30 min. Later, 5 μl sample aliquot was applied to a 230 mesh inch[−1] formvar stabilized with carbon support films, left to adsorb for 30 s, washed two times with water and then negatively stained for 40 s with 2% (w/v) phosphotungstic acid. Samples were collected and observed on an FEI T12 transmission electron microscope. Micrographs were recorded with a Serial EM software and a 2 k × 2 k Gatan CCD.

**Protein expression and purification.** The stathmin-like domain of RB3 (RB3-SLD) and TTL were overexpressed in *Escherichia coli* strain BL21(DE3)

(Novagen). Cells expressing RB3-SLD were collected by centrifugation and resuspended in loading buffer (20 mM Tris-HCl (pH 8.0), 1 mM EGTA, 2 mM dithiothreitol (DTT)) and then lysed by an ultrahigh-pressure homogenizer (JNBIO). After centrifugation, the supernatant was loaded on a Q-Sepharose FF anion exchange column (GE Healthcare), and eluted with a 0–200 mM NaCl linear gradient in 20 mM Tris-HCl and 1 mM EGTA (pH 8.0). The eluted RB3-SLD was further purified by a Superdex 200 column (GE Healthcare) in 10 mM HEPES (pH 7.2), 150 mM NaCl and 2 mM DTT. Cells expressing TTL were collected by centrifugation and resuspended in lysis buffer (50 mM Tris (pH 7.5), 1 M NaCl, 10% glycerol, 2.5 mM $MgCl_2$) supplemented with 10 mM β-mercaptoethanol and protease inhibitors (Roche), and lysed by an ultrahigh-pressure homogenizer (JNBIO). The lysate was clarified by centrifugation and loaded onto a nickel-affinity column (GE Healthcare), washed with 20 mM imidazole and eluted with 250 mM imidazole. The fractions containing TTL protein were pooled, concentrated to 1 ml and loaded onto a Superdex 200 column (GE Healthcare) for the final purification step in 20 mM Bis-Tris propane (pH 6.5), supplemented with 200 mM NaCl, 2.5 mM $MgCl_2$, 5 mM β-mercaptoethanol and 1% glycerol. Purified RB3-SLD and TTL proteins were concentrated to 10 and 20 mg ml[−1] respectively, and stored at −80 °C until use. Porcine brain tubulin was purchased from Cytoskeleton Inc. as a frozen liquid (10 mg ml[−1]) and preserved at −80 °C until use.

**Crystallization.** The proteins of tubulin (10 mg ml[−1]), TTL (20 mg ml[−1]) and RB3 (10 mg ml[−1]) were mixed with molar ratios of each component as 2:1.3:1.2 (tubulin:RB3:TTL), and incubated on ice. Later, 1 mM AMPPCP, 5 mM tyrosine and 10 mM DTT were added, and the resulting mixture concentrated to 20 mg ml[−1] at 4 °C. Crystals of T2R-TTL were obtained by the sitting-drop vapour-diffusion method at 20 °C, in a buffer consisting of 6% PEG 4000, 5% glycerol, 0.1 M MES, 30 mM $CaCl_2$ and 30 mM $MgCl_2$ (pH 6.7). Seeding method was used to obtain single crystals. Crystals appeared after incubation at 20 °C for 12 h and reached a length of 200–300 μm within 2 days. The tubulin–AJ complex was obtained by adding 0.1 μl taccalonolide AJ (dissolved in DMSO at 10 mM concentration) into the crystal-containing drop for 24 h.

**X-ray data collection and structure determination.** The crystals of tubulin–AJ complex were cryoprotected in reservoir solution supplemented with 20% glycerol. X-ray diffraction data were collected at beamline BL19U1 of National Center for Protein Sciences Shanghai at Shanghai Synchrotron Radiation Facility. Data were processed with HKL-3000 (ref. 28). The structure was determined by the molecular replacement method with the T2R-TTL structure (PDB code: 4I55) as the initial

### Table 1 | Data collection and refinement statistics.

| | Tubulin–AJ |
|---|---|
| *Data collection* | |
| Space group | $P2_12_12_1$ |
| Cell dimensions | |
| $a, b, c$ (Å) | 105.3 158.5 180.8 |
| $\alpha, \beta, \gamma$ (°) | 90.0 90.0 90.0 |
| Resolution (Å) | 50.0-2.05 (2.09-2.05)* |
| $R_{sym}$ or $R_{merge}$ | 10.8 (78.7) |
| $I/\sigma I$ | 14.6 (1.9) |
| Completeness (%) | 99.9 (99.9) |
| Redundancy | 6.7 (6.9) |
| | |
| *Refinement* | |
| Resolution (Å) | 45.2-2.05 |
| No. reflections | 188,452 |
| $R_{work}/R_{free}$ | 18.7/20.8 |
| No. of atoms | |
| Protein | 17,524 |
| Ligand/ion | 281 |
| Water | 1,547 |
| B-factors | |
| Protein | 34.1 |
| Ligand/ion | 39.9 |
| Water | 38.9 |
| R.m.s.d. | |
| Bond lengths (Å) | 0.003 |
| Bond angles (°) | 0.822 |

R.m.s., root mean squared.
*Highest resolution shell is shown within parentheses.

model. Structure refinement and model building were performed with PHENIX[29] and Coot[30]. Structure model was validated with MolProbity[31]. The final model had 98.1% of the residues in the favoured regions and no outliers had been found in the Ramachandran plot. Data collection and refinement statistics were summarized in Table 1. All structure figures were prepared with PyMol (http://www.pymol.org).

**Molecular docking simulations.** The AutoDock Vina program[32] was applied for molecular docking studies. Compound AI was prepared as pdbqt file using AutoDockTools. The X-ray crystal structure of tubulin complexed with taccalonolide AJ was used as the template. All the water molecules and solvent molecules were removed. Gasteiger–Marsili charges were added to the protein model, and nonpolar hydrogens were then merged onto their respective heavy atoms using AutoDockTools. The so-prepared tubulin structure was used as the docking template. The grid centre was set as coordinates of $[x, y, z = -1.68, -61.92, 22.73]$, and the grid size was $22\,\text{Å} \times 22\,\text{Å} \times 22\,\text{Å}$, which encompasses the taxane pocket of tubulin. The other parameters for Vina were set as default.

**Proton NMR spectrum.** All NMR spectra were recorded on a Bruker 400 (400 MHz) spectrometer. $^1$H chemical shifts were reported in $\delta$ values in p.p.m. with the deuterated solvent as the internal standard. Data are reported as follows: chemical shift, multiplicity (s = singlet, d = doublet, t = triplet, q = quartet, br = broad, m = multiplet), coupling constant (Hz) and integration.

**Gaussian calculation.** Both structures of taccalonolide AJ($\alpha$) and AJ($\beta$) were optimized in Gaussian09 (Gaussian Inc., Wallingford, CT, USA) at the B3LYP/6-31G(d) theoretical level[33,34].

**Reverse-phase HPLC.** Nucleotides GDP and GTP were purchased from Sigma-Aldrich. Tubulin protein was diluted to $2\,\text{mg ml}^{-1}$ with G-PEM (80 mM PIPES (pH 6.9), 2 mM $MgCl_2$, 0.5 mM EGTA and 1 mM GTP). Taccalonolide AJ (dissolved in DMSO) was added to tubulin protein to a final concentration of 25 μM, and the sample was incubated at 4 °C for 3 h before removing excess nucleotides by a desalting column. The control was added with same amount of DMSO. Later, the samples were concentrated to about $8\,\text{mg ml}^{-1}$ and stored at 4 °C. GTP hydrolysis was monitored by reverse-phase HPLC at different time points. The HPLC system (Waters) was used with a C18 column (Phenomenex). The solutions used were as follows: buffer A consisted of 5% acetonitrile, 5 mM tetrabutylammonium hydroxide, 25 mM $KH_2PO_4$ (pH 6.0); buffer B consisted of 60% acetonitrile, 5 mM tetrabutylammonium hydroxide, 25 mM $KH_2PO_4$ (pH 6.0). The gradient for a reverse-phase HPLC run was 0 to 70% buffer B over 45 min with a 10 μl injection of sample.

**Statistical analysis.** Statistical analysis was performed using the GraphPad Prism 6 Software. Two-tailed Student's $t$-test was used to compare differences. Significance level was set at $P < 0.05$. All values were reported as means ± s.d.

**Data availability.** Atomic coordinates and structure factors of tubulin–AJ complex have been deposited in the Protein Data Bank under accession code 5EZY. All other data are available from the corresponding authors on reasonable request.

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

## Acknowledgements

We are grateful to Dr Michel O. Steinmetz (PSI, Switzerland) for providing the plasmid of TTL. We thank the staffs from BL19U1 beamline of National Center for Protein Science Shanghai (NCPSS) at Shanghai Synchrotron Radiation Facility for assistance during data collection. We also thank Li Yang (Zeiss) and Jianhong Yang for helping use the laser confocal microscope, and Dr Dawen Niu for the discussion of the covalent reaction mechanism. Financial support for this work was provided by National Major Scientific and Technological Special Project for 'Significant New Drugs Development' (2012ZX09103301-030), National Natural Science Foundation of China

(81372822, 81501368 and 81672722) and the Fondation ARC pour la Recherche sur le Cancer.

## Author contributions

Y.W. performed expression, purification, crystallization and immunofluorescence. W.Q. and Q.C. collected diffraction data. Y.Y. and Q.C. solved the structure and conducted the structural analysis. Y.W. performed transmission electron microscopy and statistical analysis. G.L. and H.C. performed NMR work. S.-A.L. and C.W. performed GTP hydrolysis analysis. Y.W., Q.C. and J.Y. designed the experiments. Q.C. wrote the manuscript. B.G. contributed to revising the manuscript.

## Additional information

**Competing interests:** The authors declare no competing financial interests.

**Publisher's note**: 

