## [Peer review file · Nature Communications]

Reviewers' comments:

Reviewer #1 (Remarks to the Author):

This is a well written manuscript describing a series of logical and well-presented, convincing experiments that describe the binding of a newly identified class of microtubule stabilizers, the taccalonolides to tubulin. The value of microtubule stabilizers clinically and the scarcity of compound classes that have these effects makes this manuscript noteworthy and of general interest.

Two different amino acid numbering systems for β tubulin are used in the field. One is based on the actual sequence of β -tubulin and a second uses its homology to α -tubulin. The nomenclature used in the manuscript is not defined and it appears that both numbering systems are used in the text with reference to (Asp) D224 (abstract) and D226 (page 3). One numbering system needs to be used throughout and it needs to be defined in the methods. The prior publication defining the peptide of taccalonolide AJ binding, β 212-230 used the actual β -tubulin numbering system which would be consistent with D224 nomenclature. On page 4 the numbering nomenclature needs to be changed including a correction on line 115, β D22.

It would be valuable to define that the T2R-TTL crystal contains 2 $\alpha\beta$ -tubulin heterodimers that is what is defined by T2. This will clarify a point of minor confusion in the language used on page 5 where it is indicated that "AJ was found at both taxane sites". In the description of the tubulin nucleotide state it would be clearer if the authors state that AJ is bound on both β tubulin subunits in the T2R-TTL complex. The statement that AJ binds to both "taxane sites" is somewhat confusing as this could mean the pore site and the pocket site.

In looking at the chemical structure it is apparent that taccalonolide AJ has 2 epoxide groups, so it would be useful to define that the epoxide involved in the covalent binding is the C22-C23 epoxide and this should be described as such throughout the manuscript.

The chemistry defining the orientation of the C22-C23 epoxide is convincing and well documented.

The hydrogen-deuterium exchange (HDEX) experiments with taccalonolide AJ referenced used GMPCPP (non-hydrolyzable GTP analog)-stabilized microtubules, which allowed for differentiation of drug-specific effects from those associated with general microtubule stabilization. Docetaxel increased the stabilization of this M-loop even further as measured by HDEX, but AJ did not, showing a difference in the effects of these drugs on the M-loop independent of GTP hydrolysis. It would be helpful if the authors incorporated a discussion of these differences in the discussion.

Can the authors speculate on why only one of the 2 E-sites sites contains a GTP even though both β -tubulin subunits contain AJ? This manuscript demonstrates that AJ slows but does not completely inhibit GTP hydrolysis. When GTP is hydrolyzed, what is the consequence of AJ binding? Does another GTP bind before the microtubule can depolymerize? Can the microtubule depolymerize with AJ bound? Is AJ still bound when GTP hydrolysis is seen? Experiments have shown that the cellular effects of AJ persist for

extended periods of time even after the drug has been removed from the medium.

The authors seem to conclude that the slowing of GTP hydrolysis is the primary mechanism of MT stabilization by the taccalonolides, but the GTP hydrolysis is only somewhat slowed. How does this slowing of the GTP hydrolysis predict the strong inter-prot filament stability measured by HDEX and the changes in the M-loop that facilitate lateral interactions?

Three mechanisms as to how AJ stabilizes microtubules are proposed in the last paragraph of the results section and convincing data is presented for two of them, however do the authors have any evidence to support the statement that "AJ may promote tubulin polymerization by binding to soluble tubulins and locking them into the GTP status"? Is there any evidence that AJ can bind to soluble tubulin?

The authors insinuate that any taxane site agent that bound covalently would have the same effect as AJ. The same crystallographic analysis has been performed with zampanolide. The authors could thoroughly compare the binding site and allosteric effects of these two drugs, particularly with respect to their effects on the M-loop and the GTP pocket. This data would help to demonstrate whether all drugs that covalently bound within the taxane pocket had equivalent effects on the microtubule or whether there were differences between these agents.

Some unique properties of AJ compared to other classes of MSAs are the slow rate of initiating tubulin polymerization, insensitivity to cold-induced depolymerization, and their long-lasting cellular effects after drug wash-out. Are there any ideas how the proposed model can explain these interesting difference in the taccalonolides as compared to other MSAs?

The language overall is excellent, however, a few minor corrections are suggested.

Line 23: mechanisms

Line 27: the M-loop

Line 39: remove a, an add plural for structures... "Microtubules are dynamic structures"...

Line 42: heterodimers

Line 43: the α -loop

Line 60: The taccalonolides

Line 84 the M-loop:

Line 86: induces

Line 87: prone to retain GTP

Line 190: GTP-stable

Line 201: GTP-bound state

Line 225-6: this discrepancy is due to the fact that the specimens...

Line 269: add space between 1 and h

Line 274: first sentence needs to be rewritten

Reviewer #2 (Remarks to the Author):

The manuscript by Wang et al. reports the structure of tubulin complexed to a Taccanolide (AJ), determined at 2.05 Å resolution. This ligand has been studied previously and reported then (1) to favour microtubule assembly, similarly to paclitaxel, and (2) to form a covalent bond with tubulin, more specifically with a beta tubulin residue in the 212-230 range, as determined then by mass spectrometry (ref. 10 in this manuscript). The structure indeed demonstrates a covalent bond between the taccanolide and residue 226 of beta tubulin. In addition, the Taccanolide studied here seems to slightly inhibit the tubulin GTPase.

Given the effect of Taccanolide AJ on GTP hydrolysis (which is in any case marginal; see below), the main finding reported here is the structure of the complex of Taccanolide AJ with tubulin. As there are by now many isomorphous tubulin complexes structures published (see below), and since a covalent bond between tubulin and Taccanolide AJ has already been established, the findings reported here do not warrant publication in Nature Communications.

There are several remarks to be made on the findings reported:

- the resolution is good. It is in the range of the, by now many, tubulin complexes reported, mostly by the group of M. Steinmetz and A. Prota (Paul Scherrer Institute). In fact, about twenty structures have been released in the pdb.
- the establishment of a covalent bond between a ligand and tubulin has been reported previously, at least in the cases of zampanolide (a taxol site ligand too, Science (2013)) and of Pironetin (a ligand binding close to the Vinca site, J. Mol. Biol. (2016)). Therefore, though by no means general, this property is not unique to the Taccanolide studied here or to the site targeted. References to other compounds making a covalent bond with tubulin could be given in the manuscript.
- the presence of GTP at one of the hydrolysable nucleotide sites of the complex of tubulin with RB3 and tubulin tyrosine ligase is not unique to the structure reported here: it was reported previously by the group authoring this manuscript, in a control without any ligand (Yang et al. Nat. Commun. 12103 (2016)). This should be stated and discussed somewhere in this manuscript. This also questions the specificity and relevance of this observation to Taccanolide AJ.
- the effect of taccanolide AJ on GTP hydrolysis by tubulin in solution is marginal, given that there must be a significant uncertainty on measurements: without any ligand, more than 110% (!) of the hydrolysable GTP (bound to the beta subunit) is hydrolysed by tubulin in 48h (Fig. 4f). Beyond these results, GTP hydrolysis by tubulin results from collisions between tubulin molecules (see Carlier et al. BBRC, 103, pp 332-338 (1981)). It is therefore strange that a compound that favours assembly inhibits hydrolysis. The authors should provide a realistic mechanism. Given that the effect is marginal, it is not clear to this reviewer that this is of the highest priority.
- the authors should include in their discussion of the effect of Taccanolide AJ on GTP hydrolysis by tubulin the fact that, apparently, the GTP on the beta1 tubulin subunit is hydrolysed (Fig. 2a, see the pink GDP). This does not seem to be mentioned in this manuscript, other than by the colour code of Fig. 2a.

Some specific points:

- the residue of beta tubulin that reacts with Taccanolide AJ is given several numbers in the manuscript (D224, D226 and D224 again ...). This is probably related to the fact that there are mainly 2 numbering conventions for tubulin: sequential and with lined up alpha

and beta subunits. The authors should decide on one, preferably state which convention they use, and stick to it.

- the reference to Nogales et al. (Cell, 1999) for the lateral contacts made by the M loop in microtubules is not very recent. More recent ones exist that could be cited (one example is: Fourniol et al. J. Cell Biol. 191, 463 (2010)).

In addition, there are quite a few typos (or similar mistakes) in this manuscript, which should be corrected.

Reviewer #3 (Remarks to the Author):

The authors have determined the crystal structure of tubulin complexed with the stathmin-like protein RB3 plus tubulin tyrosine ligase (T2R-TTL) and also associated with a small molecule called Taccalonolide AJ, which proved to be covalently bound to β -tubulin D224. The nucleotide bound to the E-site in β -tubulin was found to be non-hydrolysed GTP and its M-loop was locked into the assembly-favourable open conformation, with part of it forming a short helix. NMR provided further information on the structure of AJ and the reaction mechanism required to form the covalent bond between the AJ and β -tubulin. The addition of AJ was also demonstrated to reduce the rate of GTP-hydrolysis by tubulin.

The results reported are novel and very interesting but I think their interpretation and the discussion require some modification. The abstract says that "the detailed molecular mechanism by which MSAs stabilize microtubules remains elusive" but actually it is not even clear that all MSAs stabilize microtubules in the same way. My other quibble with the abstract is the statement (reflecting statements throughout the paper) that AJ binding "locks the E-site into a GTP-preferred status". Free tubulin dimer always preferentially exchanges bound GDP for GTP, even in the absence of any MSA, and hydrolysis to GDP is stimulated by another tubulin dimer during assembly. The authors may argue that their studies were carried out on tubulin that was not assembled but any GTP hydrolysis in the samples presumably occurs during temporary association of different subunits.

So it would be more accurate to say that AJ binding somehow locks β -tubulin into a conformation in which GTP hydrolysis is inhibited (T7 of the next subunit in a protofilament is prevented from triggering hydrolysis because bound AJ, in some unclear way, stabilizes β -tubulin loops around the E-site GTP).

Thus, Fig. 4g is over-simplistic and would probably be best omitted.

[Even Taxol has an effect on the longitudinal bonds and this is not understood – see Elie-Caille et al (2007) Straight GDP-tubulin protofilaments form in the presence of taxol. Curr Biol. 2007 Oct 23;17(20):1765-70.]

Figures:

Fig 1: " α -tubulin antibody (blue)" green!

Fig. 2: why is the top-most nucleotide orange, not pink?

Reviewers' comments:

Reviewer #1 (Remarks to the Author):

This is a well written manuscript describing a series of logical and well-presented, convincing experiments that describe the binding of a newly identified class of microtubule stabilizers, the taccalonolides to tubulin. The value of microtubule stabilizers clinically and the scarcity of compound classes that have these effects makes this manuscript noteworthy and of general interest.

We thank the reviewer and highly appreciate the comments.

Two different amino acid numbering systems for β tubulin are used in the field. One is based on the actual sequence of β -tubulin and a second uses its homology to α -tubulin. The nomenclature used in the manuscript is not defined and it appears that both numbering systems are used in the text with reference to (Asp) D224 (abstract) and D226 (page 3). One numbering system needs to be used throughout and it needs to be defined in the methods. The prior publication defining the peptide of taccalonolide AJ binding, β 212-230 used the actual β -tubulin numbering system which would be consistent with D224 nomenclature. On page 4 the numbering nomenclature needs to be changed including a correction on line 115, β D22.

We thank the reviewer for pointing out this issue. We have solved the structure using the actual sequence numbering of β -tubulin, and then changed to the numbering system basing on its homology to α -tubulin, because we find that many other T2R-TTL structures use this homology numbering system. The structure we have deposited to PDB used the homology numbering system, so we accordingly use the homology numbering system in this manuscript. We have provided a reference (line 113-114, ref. 16) for this, and accordingly use D226 throughout this manuscript.

It would be valuable to define that the T2R-TTL crystal contains 2 $\alpha\beta$ -tubulin heterodimers that is what is defined by T2. This will clarify a point of minor confusion in the language used on page 5 where it is indicated that "AJ was found at both taxane sites". In the description of the tubulin nucleotide state it would be clearer if the authors state that AJ is bound on both β tubulin subunits in the T2R-TTL complex. The statement that AJ binds to both "taxane sites" is somewhat confusing as this could mean the pore site and the pocket site.

According to the reviewer's suggestion, we modified this part (line 106, 186-188). We thank the reviewer for his/her suggestion that makes the manuscript clearer.

In looking at the chemical structure it is apparent that taccalonolide AJ has 2 epoxide groups, so it would be useful to define that the epoxide involved in the covalent binding is the C22-C23 epoxide and this should be described as such throughout the

manuscript.

We totally agree with the reviewer. According to the reviewer's suggestion, we have used "C22-C23 epoxide" throughout the manuscript.

The chemistry defining the orientation of the C22-C23 epoxide is convincing and well documented.

We highly appreciate the comments.

The hydrogen-deuterium exchange (HDEX) experiments with taccalonolide AJ referenced used GMPCPP (non-hydrolyzable GTP analog)-stabilized microtubules, which allowed for differentiation of drug-specific effects from those associated with general microtubule stabilization. Docetaxel increased the stabilization of this M-loop even further as measured by HDEX, but AJ did not, showing a difference in the effects of these drugs on the M-loop independent of GTP hydrolysis. It would be helpful if the authors incorporated a discussion of these differences in the discussion.

We thank the reviewer for the suggestions. We have added a paragraph to address this issue in the discussion (line 238-243). Covalently bound to D226, AJ occupied a position a little more far away from the M-loop, compared to Zampa and EpoA. So AJ has a lesser effect on the M-loop than Zampa or EpoA, and the stabilizing effect of AJ on the M-loop may be beyond the detection limit of the HDXMS method.

Can the authors speculate on why only one of the 2 E-sites sites contains a GTP even though both β -tubulin subunits contain AJ? This manuscript demonstrates that AJ slows but does not completely inhibit GTP hydrolysis. When GTP is hydrolyzed, what is the consequence of AJ binding? Does another GTP bind before the microtubule can depolymerize? Can the microtubule depolymerize with AJ bound? Is AJ still bound when GTP hydrolysis is seen? Experiments have shown that the cellular effects of AJ persist for extended periods of time even after the drug has been removed from the medium.

We speculate that the E-site GTP of the middle β subunit (chain B) has already hydrolyzed to GDP before AJ binding, and could not be exchanged to GTP. We have added a paragraph to discuss this issue (line 261-273).

To form a protofilament or microtubule, subunit addition brings β -tubulin that was exposed at the plus end into contact with α -tubulin. This promotes hydrolysis of GTP bound to the now interior β -tubulin. Pi dissociates, but β -tubulin within a microtubule cannot exchange its bound GDP for GTP.

Each nucleotide in the tubulin protofilament is at an α - β interface. The inability of GTP to dissociate from the α -subunit is consistent with occlusion by a loop from the

β subunit. A similar occlusion would account for the inability of β -tubulin within a protofilament to exchange bound GDP for GTP.

In the T2R-TTL complex, the middle β subunit (chain B) is in a situation mimicking the β -tubulin within a curved protofilament, and the GDP may be not able to be released. We have used soaking method to obtain the tubulin-AJ complex, adding AJ after crystallization. If the GTP in the middle β subunit has already hydrolyzed before AJ binding, the bound AJ is not able to do anything for that. This is likely the case. Because tubulin, as a GTPase, has a low activity. Tubulin itself serves as GAP (GTPase activating protein) for β -tubulin of the adjacent dimer in a protofilament to significantly accelerate the GTP hydrolysis. However, the hydrolysis of the E-site GTP in the top β subunit (chain D) is slower, and could be released and exchanged to a new GTP after hydrolysis. The E-site GTP we have observed in the top β -subunit (chain D) has two possibilities: 1) the hydrolysis of the E-site GTP in the top β subunit (chain D) is not happened before AJ binding, and 2) the hydrolysis happened and the resulting GDP has been exchanged to a new GTP. In any case, upon AJ binding, the hydrolysis of GTP in this site will be inhibited. We think this is the reason why we only observed only one E-site GTP in our crystal structure.

The inhibiting effect of AJ on GTP hydrolysis may be underestimated in the assay reported in our manuscript, since soluble AJ has been removed after incubation. Maybe not all tubulin molecules finished the covalent reaction with AJ. We do not know any report on the microtubule depolymerization with AJ bound. We do not think AJ could inhibit the E-site GTP hydrolysis within a microtubule, since the GTPase activity of tubulin is significantly increased by assembly. Our biochemical data show that AJ inhibits the GTP hydrolysis for the soluble tubulin, and we propose that AJ may inhibit the E-site GTP hydrolysis for the first layer of GTP-cap which mimics the soluble tubulin, and thus endow the GTP-cap a prolonged GTP-status to promote the assembly of microtubule and stabilize microtubule from disassembly.

The cellular effects of AJ persist for extended periods of time even after the drug has been removed from the medium. This could be readily explained by the covalent interaction between AJ and tubulin. AJ bound to tubulin is just not able to be washed out.

The authors seem to conclude that the slowing of GTP hydrolysis is the primary mechanism of MT stabilization by the taccalonolides, but the GTP hydrolysis is only somewhat slowed. How does this slowing of the GTP hydrolysis predict the strong inter-protofilament stability measured by HDEX and the changes in the M-loop that facilitate lateral interactions?

We have proposed that AJ stabilizes microtubule by affecting both the lateral (the M-loop) and longitudinal (GTP binding site) contacts of microtubule. AJ binding induces conformational changes of the β -tubulin M-loop to facilitate establishing

lateral tubulin contacts in microtubules. This effect is directly from AJ binding, and not related to the GTP hydrolysis.

The method HDEX gives information about the solvent accessibility of various parts of the molecule, and thus the tertiary structure of the protein. The HDEX assay used microtubules as samples, while our crystal structure consists of two $\alpha\beta$ -tubulin heterodimers, an inhibiting protein RB3 and a tubulin tyrosine ligase. So the crystal structure is unlikely to show the conformational changes of the microtubule structure.

Three mechanisms as to how AJ stabilizes microtubules are proposed in the last paragraph of the results section and convincing data is presented for two of them, however do the authors have any evidence to support the statement that “AJ may promote tubulin polymerization by binding to soluble tubulins and locking them into the GTP status”? Is there any evidence that AJ can bind to soluble tubulin?

The Reverse-phase HPLC assay we have used in this manuscript to monitor the hydrolysis of E-site GTP is performed using unassembled tubulins (4°C). So the inhibiting effect we have observed is for soluble tubulins.

AJ binds to crystallized tubulin, as shown by soaking of crystals. It is not exactly soluble tubulin (it is tubulin in T2R-TTL, in a crystal), but it is not microtubule.

There is a publication (J Am Chem Soc 2011, 133:19064-7. ref. 10 in our manuscript) reported that AJ stimulated the polymerization of purified tubulin, suggesting that this class of microtubule stabilizers can interact directly with tubulin.

The authors insinuate that any taxane site agent that bound covalently would have the same effect as AJ. The same crystallographic analysis has been performed with zampanolide. The authors could thoroughly compare the binding site and allosteric effects of these two drugs, particularly with respect to their effects on the M-loop and the GTP pocket. This data would help to demonstrate whether all drugs that covalently bound within the taxane pocket had equivalent effects on the microtubule or whether there were differences between these agents.

We thank the reviewer for the suggestions. We have compared AJ with Zampa and EpoA on the M-loop (Fig. 4b), and on the GTP pocket (Fig. 4c). Zampa and EpoA occupy a closer position to the M-loop and induce it to form a longer helix (Fig. 4b). The GTP pocket of the β -tubulin in Zampa- or EpoA-tubulin complex is in GDP conformation, while the GTP pocket of the β -tubulin in AJ-tubulin complex is in GTP conformation (Fig. 4c). SO our speculation is that taxane site agents could stabilize both the lateral (the M-loop) and longitudinal (GTP binding site) contacts of microtubule. AJ has more effects on the longitudinal contacts, while Zampa and EpoA

have more effects on the lateral contacts.

Some unique properties of AJ compared to other classes of MSAs are the slow rate of initiating tubulin polymerization, insensitivity to cold-induced depolymerization, and their long-lasting cellular effects after drug wash-out. Are there any ideas how the proposed model can explain these interesting difference in the taccalonolides as compared to other MSAs?

The long-lasting cellular effects of AJ after drug wash-out may be related to the covalent interaction, hence to AJ not being washed out. However, other taccalonolides that do not contain the C22-C23 epoxide group, such as taccalonolide A, also showed the long-lasting cellular effects after drug wash-out (Risinger AL, Mooberry SL. Cell Cycle. 2011;10(13):2162-71.). It is noteworthy that the direct interaction between taccalonolide A and tubulin is even non-detectable. So we think this unique property of AJ needs further studies to elucidate the mechanism.

The slow rate of initiating tubulin polymerization could be explained by the covalent interaction between AJ and tubulin. The potency and selectivity of covalent drugs is affected by both their noncovalent affinity for their target and the second-order rate constant for covalent bond formation (described by the following equation. See "Covalent Modifiers: A Chemical Perspective on the Reactivity of α,β -Unsaturated Carbonyls with Thiols via Hetero-Michael Addition Reactions" J Med Chem. 2016). The covalent bond formation needs more time than noncovalent binding. We have added a sentence to address this issue (line 122-124).

Since the factors resulting in cold stability of microtubule are not clear, we could not explain the differences of cold stability between AJ and other MSAs. It is noteworthy that the non-hydrolyzable GTP analogs GMPPNP and GMPCPP also have the cold stable effect on microtubule. This is consistent to the inhibiting effect of AJ on E-site GTP hydrolysis.

The language overall is excellent, however, a few minor corrections are suggested.

Line 23: mechanisms

Line 27: the M-loop

Line 39: remove a, an add plural for structures..."Microtubules are dynamic structures"...

Line 42: heterodimers

Line 43: the α -loop
Line 60: The taccalonolides
Line 84 the M-loop:
Line 86: induces
Line 87: prone to retain GTP
Line 190: GTP-stable
Line 201: GTP-bound state
Line 225-6: this discrepancy is due to the fact that the specimens...
Line 269: add space between 1 and h
Line 274: first sentence needs to be rewritten

We thank the reviewer for carefully reading our manuscript. We have corrected all the grammar errors pointed out by the reviewer.

Reviewer #2 (Remarks to the Author):

The manuscript by Wang et al. reports the structure of tubulin complexed to a Taccanolide (AJ), determined at 2.05 Å resolution. This ligand has been studied previously and reported then (1) to favour microtubule assembly, similarly to paclitaxel, and (2) to form a covalent bond with tubulin, more specifically with a beta tubulin residue in the 212-230 range, as determined then by mass spectrometry (ref. 10 in this manuscript). The structure indeed demonstrates a covalent bond between the taccanolide and residue 226 of beta tubulin. In addition, the Taccanolide studied here seems to slightly inhibit the tubulin GTPase.

Given the effect of Taccanolide AJ on GTP hydrolysis (which is in any case marginal; see below), the main finding reported here is the structure of the complex of Taccanolide AJ with tubulin. As there are by now many isomorphous tubulin complexes structures published (see below), and since a covalent bond between tubulin and Taccanolide AJ has already been established, the findings reported here do not warrant publication in Nature Communications.

The most important finding in our manuscript is that taccanolide AJ could affect both the lateral (the M-loop) and longitudinal (GTP binding site) contacts of microtubule to facilitate tubulin polymerization and microtubule-stabilizing.

The hydrolysis of GTP plays a central role in microtubule assembly-disassembly kinetics. Here we for the first time provide experiment evidence to demonstrate MSA-binding could affect tubulin nucleotide state, and thus build a direct connection between them.

There are several remarks to be made on the findings reported:

- the resolution is good. It is in the range of the, by now many, tubulin complexes reported, mostly by the group of M. Steinmetz and A. Prota (Paul Scherrer Institute). In fact, about twenty structures have been released in the pdb.

We thank the reviewer for the comments. The 2.05 Å structure we report in this manuscript is among the highest resolution tubulin structures in PDB. Benefiting from the high resolution, we could determine the orientation and conformation of the ligand unambiguously, and thus exclude the common SN2 reaction mechanism for the tubulin-AJ ester linkage formation. We then propose a new reaction mechanism for it.

- the establishment of a covalent bond between a ligand and tubulin has been reported previously, at least in the cases of zampanolide (a taxol site ligand too, Science (2013)) and of Pironetin (a ligand binding close to the Vinca site, J. Mol. Biol. (2016)). Therefore, though by no means general, this property is not unique to the Taccanolide studied here or to the site targeted. References to other compounds making a covalent bond with tubulin could be given in the manuscript.

According to the reviewer's suggestion, we have added a sentence to give the references of other covalent-bound ligands (line 101-102).

- the presence of GTP at one of the hydrolysable nucleotide sites of the complex of tubulin with RB3 and tubulin tyrosine ligase is not unique to the structure reported here: it was reported previously by the group authoring this manuscript, in a control without any ligand (Yang et al. Nat. Commun. 12103 (2016)). This should be stated and discussed somewhere in this manuscript. This also questions the specificity and relevance of this observation to Taccanolide AJ.

We have mentioned in the original manuscript that the presence of GTP at E-site has been also found in a crystal due to the seeding method (line 193-194, ref. 21). Therefore we use biochemical assays to confirm that AJ inhibits the GTP hydrolysis. So our conclusion is not only based on the crystal structure.

According to the reviewer's suggestion, we have add a paragraph in discussion on this issue (line 261-273).

- the effect of taccanolide AJ on GTP hydrolysis by tubulin in solution is marginal, given that there must be a significant uncertainty on measurements: without any ligand, more than 110% (!) of the hydrolysable GTP (bound to the beta subunit) is hydrolysed by tubulin in 48h (Fig. 4f). Beyond these results, GTP hydrolysis by tubulin results from collisions between tubulin molecules (see Carlier et al. BBRC, 103, pp 332-338 (1981)). It is therefore strange that a compound that favours assembly inhibits hydrolysis. The authors should provide a realistic mechanism. Given that the effect is marginal, it is not clear to this reviewer that this is of the highest priority.

We calculate the hydrolysis rate of the E-site GTP using this equation: $2 \cdot \text{GDP} / (\text{GDP} + \text{GTP})$. The amount of GDP or GTP is measured by calculating the peak

area of GDP or GTP on HPLC. There is an assumption that the GTP on α -tubulin N-site does not hydrolyze. However, the purified tubulin protein is not stable in solution. During the incubation at 4°C, it is very likely that some tubulin heterodimers became denatured. Then the GTP bound to the N-site will be released and may be exchanged to the E-site of another tubulin heterodimer, and undergo hydrolysis. So we think it is reasonable that after 48 hours incubation at 4°C, the apparent hydrolysis rate of the E-site GTP got a value above 100%.

We do not think AJ could inhibit the E-site GTP hydrolysis within a microtubule, since the GTPase activity of tubulin is significantly increased by assembly. As mentioned in the manuscript, we proposed AJ inhibit the E-site GTP hydrolysis in the first layer of GTP-cap, and endows the GTP-cap a prolonged GTP-status to promote the assembly of microtubule and stabilize microtubule from disassembly. AJ may also increase the concentration of the soluble GTP-status tubulin.

- the authors should include in their discussion of the effect of Tacanolide AJ on GTP hydrolysis by tubulin the fact that, apparently, the GTP on the beta1 tubulin subunit is hydrolysed (Fig. 2a, see the pink GDP). This does not seem to be mentioned in this manuscript, other than by the colour code of Fig. 2a.

According to the reviewer's suggestion, we have added a paragraph to discuss this issue (line 261-273).

To form a protofilament or microtubule, subunit addition brings β -tubulin that was exposed at the plus end into contact with α -tubulin. This promotes hydrolysis of GTP bound to the now interior β -tubulin. Pi dissociates, but β -tubulin within a microtubule cannot exchange its bound GDP for GTP.

Each nucleotide in the tubulin protofilament is at an α - β interface. The inability of GTP to dissociate from the α -subunit is consistent with occlusion by a loop from the β subunit. A similar occlusion would account for the inability of β -tubulin within a protofilament to exchange bound GDP for GTP.

In the T2R-TTL complex, the middle β subunit (chain B) is in a situation mimicking the β -tubulin within a curved protofilament, and the GDP may be not able to be released. We have used soaking method to obtain the tubulin-AJ complex, adding AJ after crystallization. If the GTP in the middle β subunit has already hydrolyzed before AJ binding, the bound AJ is not able to do anything for that. This is likely the case. Because tubulin, as a GTPase, has a low activity. Tubulin itself serves as GAP (GTPase activating protein) for β -tubulin of the adjacent dimer in a protofilament to significantly accelerate the GTP hydrolysis. However, the hydrolysis of the E-site GTP in the top β subunit (chain D) is slower, and could be released and exchanged to a new GTP after hydrolysis. The E-site GTP we have observed in the top β -subunit (chain D) has two possibilities: 1) the hydrolysis of the E-site GTP in the top β subunit

(chain D) is not happened before AJ binding, and 2) the hydrolysis happened and the resulting GDP has been exchanged to a new GTP. In any case, upon AJ binding, the hydrolysis of GTP in this site will be inhibited. We think this is the reason why we only observed only one E-site GTP in our crystal structure.

Some specific points:

- the residue of beta tubulin that reacts with Tacanolide AJ is given several numbers in the manuscript (D224, D226 and D224 again ...). This is probably related to the fact that there are mainly 2 numbering conventions for tubulin: sequential and with lined up alpha and beta subunits. The authors should decide on one, preferably state which convention they use, and stick to it.

We thank the reviewer for pointing out this issue. We have solved the structure using the actual sequence numbering of β -tubulin, and then changed to the numbering system basing on its homology to α -tubulin, because we found that many other T2R-TTL structures use this homology numbering system. The structure we have deposited to PDB used the homology numbering system, so we consistently use the homology numbering system in this manuscript. We have provided a reference (line 113-114, ref 16) for this, and accordingly use D226 throughout this manuscript.

- the reference to Nogales et al. (Cell, 1999) for the lateral contacts made by the M loop in microtubules is not very recent. More recent ones exist that could be cited (one example is: Fourniol et al. J. Cell Biol. 191, 463 (2010)).

We thank the reviewer for kindly providing this information. We have updated this reference accordingly.

In addition, there are quite a few typos (or similar mistakes) in this manuscript, which should be corrected.

We have carefully checked the manuscript and corrected the typos and grammar errors.

Reviewer #3 (Remarks to the Author):

The authors have determined the crystal structure of tubulin complexed with the stathmin-like protein RB3 plus tubulin tyrosine ligase (T2R-TTL) and also associated with a small molecule called Tacalonolide AJ, which proved to be covalently bound to β -tubulin D224. The nucleotide bound to the E-site in β -tubulin was found to be non-hydrolysed GTP and its M-loop was locked into the assembly-favourable open conformation, with part of it forming a short helix. NMR provided further information on the structure of AJ and the reaction mechanism required to form the covalent bond between the AJ and β -tubulin. The addition of AJ was also demonstrated to reduce the rate of GTP-hydrolysis by tubulin.

We thank the reviewer for the comments.

The results reported are novel and very interesting but I think their interpretation and the discussion require some modification. The abstract says that “the detailed molecular mechanism by which MSAs stabilize microtubules remains elusive” but actually it is not even clear that all MSAs stabilize microtubules in the same way.

We totally agree with the reviewer that it is not clear that all MSAs stabilize microtubules in the same way. Accordingly, We have deleted the word "detailed" from this sentence. Due to the length limit of abstract (less than 150 words), we could not say more in the abstract.

My other quibble with the abstract is the statement (reflecting statements throughout the paper) that AJ binding “locks the E-site into a GTP-preferred status”. Free tubulin dimer always preferentially exchanges bound GDP for GTP, even in the absence of any MSA, and hydrolysis to GDP is stimulated by another tubulin dimer during assembly. The authors may argue that their studies were carried out on tubulin that was not assembled but any GTP hydrolysis in the samples presumably occurs during temporary association of different subunits.

So it would be more accurate to say that AJ binding somehow locks β -tubulin into a conformation in which GTP hydrolysis is inhibited (T7 of the next subunit in a protofilament is prevented from triggering hydrolysis because bound AJ, in some unclear way, stabilizes β -tubulin loops around the E-site GTP).

Thus, Fig. 4g is over-simplistic and would probably be best omitted.

[Even Taxol has an effect on the longitudinal bonds and this is not understood – see Elie-Caille et al (2007) Straight GDP-tubulin protofilaments form in the presence of taxol. *Curr Biol.* 2007 Oct 23;17(20):1765-70.]

We thank the reviewer for the comments. This is the most important point we want to answer in this manuscript. As the reviewer has pointed out, tubulin itself serves as GAP (GTPase activating protein) for β -tubulin of the adjacent dimer in a protofilament to significantly accelerate the GTP hydrolysis. However, the soluble tubulin has a low GTPase activity. In the biochemical assay we reported in this manuscript, we use a relatively high concentration of tubulin (8 mg/ml) to increase temporary association of different tubulin subunits, and thus accelerate the GTP hydrolysis. Our results showed that AJ could significantly inhibit the GTP hydrolysis of the soluble tubulins. We do not think AJ could inhibit the E-site GTP hydrolysis within a microtubule, since the GTPase activity of tubulin is significantly increased by assembly. We think AJ inhibit the E-site GTP hydrolysis in the first layer of GTP-cap, and endows the GTP-cap a prolonged GTP-status to promote the assembly of microtubule and stabilize microtubule from disassembly. AJ may also increase the concentration of the soluble GTP-status tubulin.

By saying "locks the E-site into a GTP-preferred status", we mean AJ binding could maintain the E-site a conformation in which GTP hydrolysis is inhibited, and thus endow the E-site a prolonged GTP status.

As the reviewer has pointed out, taxol also has an effect on the longitudinal bonds. Consistently, a recent EM study (Cell, 2014, ref. 20 in this manuscript) showed that taxol allosterically affected remodeling of the longitudinal interdimer interface. Our results showed that tacanolide AJ could affect both the lateral (the M-loop) and longitudinal (GTP binding site) contacts of microtubule to facilitate tubulin polymerization and microtubule-stabilizing. The hydrolysis of GTP plays a central role in microtubule assembly-disassembly kinetics. Here we for the first time provide experiment evidence to demonstrate MSA-binding could affect tubulin nucleotide state, and thus build a direct connection between them.

Fig. 4g is a summary of the microtubule-stabilizing mechanism of AJ: 1) the M-loop of β subunit undergoes a closed-to-open and loop-to-helix conformational change; 2) AJ is shown as a lock, and the β subunit interface is locked into a conformation in which GTP hydrolysis is inhibited (this conformational change is represented by the shape of β subunit interface changing from wave to sawtooth). This figure visually depicts the mechanism we have proposed in this manuscript, so we prefer to keep it.

Figures:

Fig 1: " α -tubulin antibody (blue)" green!

We thank the reviewer for pointing out this issue. We have corrected it.

Fig. 2: why is the top-most nucleotide orange, not pink?

Because the top-most nucleotide is GTP, not GDP. This finding triggers us to propose AJ could inhibit the hydrolysis of the E-site GTP, and we then use biochemical assays to confirm it.

REVIEWERS' COMMENTS:

Reviewer #1 (Remarks to the Author):

This is an improved manuscript, the authors addressed each of the concerns raised during the prior review and the changes have strengthened the manuscript. Only a few comments.

On page 3 line, 76 the authors state that the prior studies suggested that the taccalonolides AF and AJ bind covalently and in viewing that paper the results were unequivocal so it would be more appropriate use the word demonstrated and not suggested.

Page 3 line 90. The studies do provide interesting insights into the microtubule stabilizing mechanism of the taccalonolides AJ as compared to other MSAs. This should be made clear. More recent global studies on mechanisms of action of different MSAs was recently published (Kellogg et al J. Mol. Biol. 429, 633-646, 2017).

Page 3, line 102, the addition of which histidine (lowercase H) 229 and cysteine would be helpful.

On page 4, line 122-124. The authors speculate that the covalent binding of the taccalonolides AF and AJ explains their slow rate of tubulin polymerization, yet zampanolide, another microtubule stabilizer that binds covalently causes an immediate increase in turbidity in a tubulin polymerization assay (Field et al., 2009 J Med. Chem 7328-7332). Thus, a covalent binding reaction cannot explain the slow rate of tubulin polymerization seen with the taccalonolide AJ. Based on the authors' studies it seems much more reasonable to suggest the slow reaction with tubulin is related instead to the proposed reaction mechanism shown in Fig 3. This is much more compelling and this explanation of the activation followed by two interesterification processes is much more logical argument for the time delay observed with AJ. The slowness of the covalent mechanism, in and of itself, is not consistent with other covalent binding MSAs.

Page 4, line 134-136. The α -configuration of the epoxide group on AJ shown in supplemental data is consistent with a recent paper that showed up in a PubMed search of the taccalonolides that also showed the α -configuration. Risinger et al., J. Nat. Prod. 80, 409-414, 2017.

Minor corrections,

page 2, line 49, change binds to bind agents bind

page 2, line 60, change to: isolated from plants of the genus Tacca

page 3, line 65, resistance protein 7 rather than resistant protein 7

page 4, line 111 perhaps within the taxane site is better than bound to

page 4, line 147 epoxide moiety rather than ring

page 5, line 158, to form by an SN2 reaction.

Page 6, line 201, microtubules (plural)

Page 6, line 219, The taccalonolides...

Page 7, line 281, to MSAs that bind within the taxane site (since there are many poses possible)

Reviewer #3 (Remarks to the Author):

The authors have responded rereasonably to comments on the previous version of this manuscript. I see no reason why it should not now be published.

REVIEWERS' COMMENTS:

Reviewer #1 (Remarks to the Author):

This is an improved manuscript, the authors addressed each of the concerns raised during the prior review and the changes have strengthened the manuscript. Only a few comments.

We thank the reviewer and appreciate the comments.

On page 3 line, 76 the authors state that the prior studies suggested that the taccalonolides AF and AJ bind covalently and in viewing that paper the results were unequivocal so it would be more appropriate use the word demonstrated and not suggested.

According to the reviewer's suggestion, we have changed the word "suggested" to "demonstrated".

Page 3 line 90. The studies do provide interesting insights into the microtubule stabilizing mechanism of the taccalonolides AJ as compared to other MSAs. This should be made clear. More recent global studies on mechanisms of action of different MSAs was recently published (Kellogg et al J. Mol. Biol. 429, 633-646, 2017).

According to the reviewer's suggestion, we have changed the sentence to "Our studies provide insights into the microtubule-stabilizing mechanism of the taccalonolides AJ as compared to other MSAs ...".

Page 3, line 102, the addition of which histidine (lowercase H) 229 and cysteine would be helpful.

According to the reviewer's suggestion, we have added the residue number to specify the histidine and cysteine (β H229 and α C316).

On page 4, line 122-124. The authors speculate that the covalent binding of the taccalonolides AF and AJ explains their slow rate of tubulin polymerization, yet zampanolide, another microtubule stabilizer that binds covalently causes an immediate increase in turbidity in a tubulin polymerization assay (Field et al., 2009 J Med. Chem 7328-7332). Thus, a covalent binding reaction cannot explain the slow rate of tubulin polymerization seen with the taccalonolide AJ. Based on the authors' studies it seems much more reasonable to suggest the slow reaction with tubulin is related instead to the proposed reaction mechanism shown in Fig 3. This is much more compelling and this explanation of the activation followed by two interesterification processes is much more logical argument for the time delay observed with AJ. The slowness of the covalent mechanism, in and of itself, is not

consistent with other covalent binding MSAs.

We thank the reviewer for the suggestions. Accordingly, we explained the time delay observed with AJ by the reaction mechanism we have proposed in Fig. 3, and moved this to the "Reaction mechanism of AJ covalent binding to β -tubulin" part (Page 5, line 160-162).

Page 4, line 134-136. The α -configuration of the epoxide group on AJ shown in supplemental data is consistent with a recent paper that showed up in a PubMed search of the taccalonolides that also showed the α -configuration. Risinger et al., J. Nat. Prod. 80, 409-414, 2017.

The configuration of the epoxide group on AJ is proposed to be α since the original report of AJ (Cancer Res 63, 3211-20 (2003)). We re-evaluate this issue because this α -configuration is not consistent with the SN2 reaction mechanism. In this manuscript, we confirm that the configuration of the epoxide group on AJ is α , and thus we propose a new reaction mechanism for the tubulin-AJ ester linkage (Fig. 3).

Minor corrections,

page 2, line 49, change binds to bind agents bind

page 2, line 60, change to: isolated from plants of the genus Tacca

page 3, line 65, resistance protein 7 rather than resistant protein 7

page 4, line 111 perhaps within the taxane site is better than bound to

page 4, line 147 epoxide moiety rather than ring

page 5, line 158, to form by an SN2 reaction.

Page 6, line 201, microtubules (plural)

Page 6, line 219, The taccalonolides....

Page 7, line 281, to MSAs that bind within the taxane site (since there are many poses possible)

We thank the reviewer for carefully reading our manuscript. We have corrected all the grammar errors pointed out by the reviewer.

Reviewer #3 (Remarks to the Author):

The authors have responded rereasonably to comments on the previous version of this manuscript. I see no reason why it should not now be published.

We thank the reviewer and appreciate the comments.